# Classification of HEp-2 Staining Pattern Images Using Adapted Multilayer Perceptron Neural Network-Based Intra-Class Variation of Cell Shape

**DOI:** 10.3390/s23042195

**Published:** 2023-02-15

**Authors:** Khamael Al-Dulaimi, Jasmine Banks, Aiman Al-Sabaawi, Kien Nguyen, Vinod Chandran, Inmaculada Tomeo-Reyes

**Affiliations:** 1School of Electrical Engineering and Robotics, Queensland University of Technology (QUT), Brisbane, QLD 4000, Australia; 2School of Computer Science, Queensland University of Technology (QUT), Brisbane, QLD 4000, Australia; 3School of Electrical Engineering and Telecommunications, University of New South Wales, Sydney, NSW 2052, Australia

**Keywords:** classification, HEp-2 staining pattern image, cell shape, multilayer perceptron neural network, intra-class variation

## Abstract

There exists a growing interest from the clinical practice research communities in the development of methods to automate HEp-2 stained cells classification procedure from histopathological images. Challenges faced by these methods include variations in cell densities and cell patterns, overfitting of features, large-scale data volume and stained cells. In this paper, a multi-class multilayer perceptron technique is adapted by adding a new hidden layer to calculate the variation in the mean, scale, kurtosis and skewness of higher order spectra features of the cell shape information. The adapted technique is then jointly trained and the probability of classification calculated using a Softmax activation function. This method is proposed to address overfitting, stained and large-scale data volume problems, and classify HEp-2 staining cells into six classes. An extensive experimental analysis is studied to verify the results of the proposed method. The technique has been trained and tested on the dataset from ICPR-2014 and ICPR-2016 competitions using the Task-1. The experimental results have shown that the proposed model achieved higher accuracy of 90.3% (with data augmentation) than of 87.5% (with no data augmentation). In addition, the proposed framework is compared with existing methods, as well as, the results of methods using in ICPR2014 and ICPR2016 competitions.The results demonstrate that our proposed method effectively outperforms recent methods.

## 1. Introduction

The analysis and classification of HEp-2 cell staining patterns of histopathological images are important processes in diagnosing autoimmune diseases [1]. Computer-Aided Diagnosis (CAD) techniques have been introduced to reduce the issues of manual annotation and classification [2]. This can help to reduce the error rate of decisions during the stage of diagnosis disease [3]. A CAD system can also be used as an aid in the training and education of specialised medical personnel. Recently, DL techniques have been widely used in biomedical, biological and medical image analysis applications for CAD system development to support physician and pathologists in making an effective and accurate decision of diagnosing disease [4]. Since several international contests have been conducted in the last few years, many research studies have been proposed in relation to automatic pattern recognition and classification of HEp-2 staining microscopic images at cell and specimen levels [5,6]. The HEp-2 tests produces variety of staining patterns; therefore, the structure of a cell includes the cytoplasm, nucleus, chromosomes, and nucleoli. These types are different in terms of their number, shape, location, and size of the cell. This can help an expert to distinguish among staining patterns to differentiate autoimmune diseases [7,8].

In contrast, existing CAD systems involve five fundamental aspects [9]: (i) image acquisition; (ii) enhancement; (iii) segmentation (iv) extraction and selection features; and (v) design of classification models. The CAD procedure and its performance depend on these fundamental aspects while the classification performance is influenced by the segmentation and feature extraction processes. Recently, automatic HEp-2 cell classification based on the extraction different features has become an active area of histopathological imaging research. Features that have been used to classify HEp-2 cells, consist of: (i) geometrical features, including shape index histogram with donut-shaped spatial pooling [10]; (ii) texture features, including analysis of morphological and textural feature [11]; and (iii) colour feature, including grayscale representation of HEp-2 cell [12]. The classification methods of histopathological images face certain challenges due to the variations in cell patterns, using different stains, variations in shape due to transformation operations, and time-consuming. In addition, a single specimen often contains cells with different patterns. There are also other issues, such as large numbers of cells per image, poor-quality of images, and clustering of cells. In-homogeneous illumination of images can cause huge intra-class variations, which negatively impact HEp-2 cell recognition and classification. Another issue is the overfitting of features used for classification, due to the very high dimensionality of features compared with the relatively small number of images for model training. HEp-2 cell classification is therefore a crucial processing step.

This work is inspired by recent advances in research into feature representation schemes and multi-class classification. It is also motivated by the development of machine learning and DL techniques addressing the problems of HEp-2 cell classification in CAD system, especially overfitting of features, stained images, intra-class variation and large-scale data volume problems. In this paper, a Multilayer Perceptron (MLP) neural network is adapted by using two hidden layers to classify the Higher Order Spectra (HOS) features into six classes based on the shapes of the HEp-2 staining patterns at cell level. The first new hidden layer includes the L-moments functions, which calculate the variation in the mean, scale, skewness and kurtosis of features vector. The second layer is calculated based on a Softmax activation function, which returns the probabilities of each class, where the target class has the highest probability. The proposed method is compared with current techniques and the results of classification competitions hosted at ICPR2014 and ICPR2016. The contribution of the work is that: (i) provides a robust technique for addressing the challenges of HEp-2 cell classification; (ii) unpublished method for analysis and classification of histopathological HEp-2 cell images using DL technique; (iii) produces an accurate recognition and classification processes of HEp-2 staining pattern at cell level for increasing the accuracy of CAD system at early diagnosing stages; (iv) provides a method that can recognize six classes of Hep-2 cell “(homogeneous, speckled, nucleolar, centromere, nuclear membrane, and golgi)”; and (v) provides high performance and benchmarking for classification of Hep-2 cell comparing with state-of-the-art methods.

This organisation of paper is as: Section 2 includes the explanation of the related work; Section 3 explains the proposed method; experiments and data description is detailed in Section 4; result evaluation is discussed in Section 5; and Section 6 summarizes the conclusion of this paper.

## 2. Related Work

Different machine learning and Deep Learning (DL) techniques have been proposed widely in different fields, including biomedical and medical images, remote sensing, biometric recognition, health informatics applications and so more. For instance, diagnosis of breast cancer from histopathological images [13], determination of Autoantibodies against HEp-2 cells (BCA) [14], medical image analysis [15], detection of COVID-19 from chest x-ray images [16], detection of biomedical imaging [17] white blood cell segmentation [18,19], classification of white blood cells [20,21], and HEp-2 cell segmentation from histopathological images [22]. Problems related to the classification of HEp-2 cell staining patterns from histopathological images have attracted the attention of many researchers in terms of benchmarking and comparison, and particularly the contests held at international conferences, such as pattern recognition ICPR2012, ICPR2014 and ICPR2016 as well as International Conference Image Processing (ICIP) [23].

Machine learning methods have been previously proposed to classify HEp-2 cells. The research of [24] used vector of hierarchically method for HEp-2 classification. A K-NN classifier had been been used [25] to classify HEp-cells. A linear Support Vector Machine (SVM) learning strategy and majority was proposed in [26] for specimen-HEp-2 level classification based on three types of features SID, SIFT and SURF [27]. A Gaussian mixture model has proposed in [28] for HEp-2 cell-level classification using different groups of features, including texture, statistical, spectral, binary, and intensity features. In [29], they used “region-level classification and majority voting for classification method”. They extract several sets of features for classification, including the adjacent local binary patterns, the covariance of the intensity, the local projection coefficients, and the morphological features. Recently, DL techniques have been proposed and merged with different techniques to classify HEp-2 cells. In [30], QRFIRMLP technique was used for the recognition of HEp-2 cells using a class of temporal processing neural networks based on finite impulse response filters. In [31], a DCAE technique has been proposed to perform feature extraction via an encoding–decoding scheme. They are automatically discriminated, during the feature learning process, and representations produce by the DCAE. CNNs, pattern histograms and linear SVMs were proposed in [32] for the classification of HEp-2 specimen patterns. In [33], deep CNNs were used to extract features directly from the pixel values of the cell image in a hierarchical way, without requirement to resort to hand-crafted features in order to classify HEp-2 cells [33]. The results showed that DL methods were successed against various large-scale benchmarks for the classification of HEp-2 specimen pattern images. However, all above methods still have challenges affecting the classification process and computation processing time, for example: (i) these methods lose discriminated information when using a fixed input image size, specifically with deformed edge in the segmented images; (ii) building a visual dictionary is complicated process such using the method in [32], especially when using a large number of training images in the model due to using data augmentation; and (iii) the segmentation process may affect negatively the performance of the classification process.

## 3. Proposed Method

The proposed method consists of various processes for classifying HEp-2 cell level patterns, as shown in Figure 1.

### 3.1. Segmentation of HEp-2 Cell Staining Images

HEp-2 cells were segmented by [22,34]. The segmentation method was proposed the level set method via Geometric Active Contours (GACs) to detect the HEp-2 cells shape information from poor quality microscope images.

### 3.2. Feature Extraction

Feature extraction is an important process in the automatic classification of the HEp-2 cells. To achieve the final feature extraction task, three steps are carried out from [20]. The steps are:(1)Radon projection: We use a Radon projection to convert a two dimension image into one dimension vector. In this paper, R[θ] is calculated using the procedure from research of [20].(2)Bispectrum: is the product of the Fourier coefficients at component frequencies [35]. The bispectrum *S* in the frequency domain is then written:
(1)$$S\left(k_{1},k_{2}\right)=F\left(k_{1}\right)F\left(k_{2}\right)*F\left(k_{1}+k_{2}\right)$$“where the Fourier transform is represented by F(k) of *R* at each θ in the range [0,180] degrees. $k_{1}$ and $k_{2}$ are the normalised frequencies divided by one and half of the sampling frequency, and are in the range [0, 1] ” [35].(3)Bispectral invariants: “is a complex value that contains the information of the integrated bispectrum along a straight line *a* which gives a slope. *a* means the bispectral invariant feature of each θ, and is known as the phase.” A more detailed explanation of this method can be found in [20,35].

### 3.3. MultiLayer Perceptron

A MLP is one of a supervised DL technique that can learn $f\left(\cdot \right):R^{m}\to R^{o}$ function. It is trained on a dataset, where the number of dimensions of the input is represented by *m* and the number of dimensions of the output is represented by *o*. Suppose a set of features $X=x_{1},x_{2},\ldots ,x_{m}$ is given where the output is *y*, it can learn a non-linear function approximation for either classification or regression. In this paper, the hidden layers are based on two functions: (a) L-Moment measuring and (b) Softmax Activation Function, as described below:

#### 3.3.1. L-Moment Measuring

The L-moment measures location, while the L-moment ratio measures the scale, skewness and kurtosis. A hidden layer in the MLP classifier is L-moments that calculation is used from [21]. Where the data $H{\left(a\right)}_{\theta }$ are in ascending order, and *a* is the size of the individual projections (the length of the vector used to collect the results of each line integral). L-moment is used H(a) which indicated bispectral invariant feature vector, “L-Mean, L-Scale, L-Skewness and L-Kurtosis” are written as follows:-L-Mean which considers location features of cell, where LM=L1, and $L1=\beta_{0}$,-L-Scale which measures variation in scaling of the cell, where LS=L2/L1, and $L2=2\beta_{1}-\beta_{0}$,-L-Skewness which measures variation in concavity of cell, where LSK=L3/L2, and $L3=6\beta_{2}-6\beta_{1}+\beta_{0}$,-L-Kurtosis which measures variation in sharpness of cell, where LK=L4/L2, and $L4=20\beta_{3}-30\beta_{2}+12\beta_{1}-\beta_{0}$.

$\beta_{0}$, $\beta_{1}$, $\beta_{2}$ and $\beta_{3}$ are written in Equations (4)–(7), where *n* is θ from 0 to 90 to measure the variation of cell shape
(2)$$\begin{array}{cc} \beta_{0} & =\frac{1}{n}\sum_{j=1}^{n}H_{j} \end{array}$$
(3)$$\begin{array}{cc} \beta_{1} & =\frac{1}{n}\sum_{j=2}^{n}H_{j}\left[\left(j-1\right)/\left(n-1\right)\right] \end{array}$$
(4)$$\begin{array}{cc} \beta_{2} & =\frac{1}{n}\sum_{j=3}^{n}H_{j}\left[\left(j-1\right)\left(j-2\right)/\left(n-1\right)\left(n-2\right)\right] \end{array}$$
(5)$$\begin{array}{cc} \beta_{3} & =\frac{1}{n}\sum_{j=4}^{n}H_{j}\left[\left(j-1\right)\left(j-2\right)\left(j-3\right)/\left(n-1\right)\left(n-2\right)\left(n-3\right)\right] \end{array}$$

The first hidden layer applies L-moments function, which calculates “ L-mean, L-scale, L-skewness and L-kurtosis”. The hidden layers can perform nonlinear transformations of the inputs entered into the network. Each hidden layer function is specialized to produce good features by calculating L-moments.

#### 3.3.2. Softmax Activation Function

For multiple-classes, *x* is a hidden vector of features of LM,LS,LKS,LK containing *K* classes. It can pass through the function instead of passing through the logistic function. The node that has the highest value can be the input member of its class. The Softmax Activation Function corresponds to the neural network output representing the probability value that the input belongs to the certain class. This is written as:(6)$$P\left(y=j|z_{i}\right)=\mathrm{softmax}{\left(z\right)}_{i}=\frac{\exp(z_{i})}{\sum_{l=1}^{k}\exp\left(z_{l}\right)}$$
where *z* is defined in follow equation:(7)$$z=w_{1}x_{1}+\ldots +w_{m}x_{m}+b=\sum_{l=1}^{m}w_{l}x_{l}+b=w^{T}x+b$$
where *w* is a weight vector, *x* is a feature vector of one training sample, and *b* is a bias unit. The role of Softmax function “ is computing the probability that this training sample $x_{i}$ can belong to class *j* by giving the weight and net input $z_{i}$. The probability “$p(y=j|x^{\left(i\right)};w_{j})$ for each class label in j=1,…,k can be computed. ”

#### 3.3.3. Cross-Entropy Function

The mean squared error cost function can be used here to optimise the cost when a Softmax activation function is used at the output layer. The cross-entropy (C) in multi-class classification problems, is known to outperform the gradient descent function and is computed as follows: First, we need to define a cost function J(·) that represents the average of all cross-entropies over the *n* training samples that requires to minimise:(8)$$J\left(W;b\right)=\frac{1}{n}\sum_{i=1}^{n}C\left(Y_{i},O_{i}\right),$$
(9)$$C\left(Y_{i},O_{i}\right)=-\sum_{m}Y_{i}\cdot \log\left(O_{i}\right)$$
*Y* is the “target”, i.e., the true class labels, and the *O* is the probability computed via Softmax (output), but it is not the predicted class label. The derivative cost is computed as below equation in order to train our Softmax model and determine the weight coefficients via a gradient descent method:(10)$$\nabla w_{j}J\left(W;b\right)=\frac{1}{n}\sum_{i=0}^{n}\left[x^{\left(i\right)}\left(O_{i}-Y_{i}\right)\right]$$

Then, the cost derivative is used to update the weights w in opposite direction of the cost gradient with learning rate η for each class j∈{0,1,…,k} [36]:(11)$$w_{j}:=w_{j}-\eta \nabla w_{j}J\left(W;b\right)$$
where $w_{j}$ is the weight vector of the class y=j. Then, the bias units are updated as Equation (Equation 12):(12)$$b_{j}:=b_{j}-\eta \left[\frac{1}{n}\sum_{i=0}^{n}\left(O_{i}-T_{i}\right)\right].$$

We add an additional bias and a regularisation term “to reduce the variance of the model and decrease the degree of overfitting”, such as the L2 term with the regularisation parameter λ: $L2:\frac{\lambda }{2}{\left||w|\right|}_{2}^{2}$ in Equation (Equation 13) where
(13)$${\left||w|\right|}_{2}^{2}=\sum_{l=0}^{m}\sum_{j=0}^{k}w_{i,j}$$

The cost function becomes:(14)$$J\left(W;b\right)=\frac{1}{n}\sum_{i=1}^{n}C\left(T_{i},O_{i}\right)+\frac{\lambda }{2}{\left||w|\right|}_{2}^{2}$$

The “regularised” weight update is defined as:(15)$$w_{j}:=w_{j}-\eta \left[\nabla w_{j}J\left(W\right)+\lambda w_{j}\right].$$

The regularization parameter λ considers an input to our model, which reduces overfitting, by reducing the variance of our estimated regression parameters. Increasing λ results with less overfitting, but also has highest bias. One approach is randomly sub-sampled data number of times and looked at the variation in our estimate. Then, repeating the process for a slightly higher value of λ may affect the variability of our estimate model. We used a small value that can help achieving comparable regularization on the whole data set. In Figure 2, the leftmost layer, which is known as the input layer, consists of a set of neurons representing the input features $x=x_{1},x_{2},\ldots ,x_{m}$ which represent the HOS features H(a). The hidden layer consists of two layers which transform each neuron in the values from the previous layer, using a weighted linear summation $w_{1}x_{1}+w_{2}x_{2}+\ldots +w_{m}x_{m}$: The first hidden layer applies L-moments function, which calculates L-mean, L-scale, L-skewness and L-kurtosis. This is followed by a Softmax activation function $R\to Y^{6}$. The softmax function calculates the probabilities variation of mean, scale, skewness and kurtosis of each target class over all possible target classes. The range value of Softmax will be from 0 to 1, and the sum of all the probabilities value should be equal to 1. For multi-classification model, the softmax function returns the value of probabilities of each class, and the target class will have the highest value probability. The output layer receives the values from the last hidden layer and transforms them into output values [37].

## 4. Experimental Analysis

Our experimental results are analyzed as follows:

### 4.1. Description of Dataset

The data set for Task-1 collected at the Sullivan Nicolaides pathology (SNP) laboratory from evaluations of positive serum samples for 419 patients. The dataset includes two classes: positive cells and intermediate cells, as shown Figure 3.

“The six patterns of cell staining in the dataset are: homogeneous (Hm), speckled (Sp), nucleolar (Nu), centromere (Cn), nuclear membrane (Nm), and Golgi (Gl), as shown in Figure 4. In the cell level classification of staining patterns, there are a total of 13,596 categorised cell images which each class has: homogeneous (2494), speckled (2831), nucleolar (2598), centromere (2741), Golgi (724) and nuclear membrane (2208).” The training set is augmented by rotation with an angle step of (90) and (30) to increase the number of images in the dataset, and to take the overfitting issue into consideration. The trained images are cropped to a size of 50×50 in order to standardise the size of the images before the rotation process is applied.

### 4.2. Implementation of Proposed Method

In this paper, classifying HEp-2 cell is implemented using MATLAB 2022b, and the process is illustrated in Figure 1. The proposed technique is used 80% of the dataset for the training set (10,833 images) and the remaining 20% for the testing set (2717 images). The steps of the proposed method are:1Pre-processing and segmentation:Pre-processing is done by adjusting the intensity of image for increasing the contrast of the image. A level set method via edge-based GACs is then applied to detect the HEp-2 cell shape information from original microscope images used by [22].2Feature extraction using HOS:The HOS technique is applied to the results of segmentation to extract features. The segmented HEp-2 cell image has been then converted to a set of 1D vectors using the “MATLAB Radon projection function”. This function can produce a Radon vector *R* for each angel from 0 to 180 degrees. A total of 256 features have been extracted, and the length of FFT used for each Radon projection is 1024. Finally, we obtained a set of 23,040 features for each image.3MLP using Softmax regression via gradient descent:For the work presented in this paper, the neural network is implemented using Python 3.5.8. The MLP classifier model has four layers. Firstly, we encode the class labels into a certain format. One-hot encoding is applied, in which a sample belonging to Class-1 has the 1 value in the first cell of its row; a sample belonging to Class-2 has the 1 value in the second cell of its row, and so on. The input layer is a vector of (23,040) features multiplying by training images size of 10,833 × 23,040. Then, we initialise the parameter of weight matrix size of 10,833 × 23,040 × 6 (one column for each class and one row feature), where *k* represents four weights for each node. For example, the first row the matrix of dimensional weights is [0.1 0.2 0.3 0.4 0.5 0.6]. We construct a neural network with two hidden layers. The first hidden layer is calculated by summing the L-moments function, which includes the “ L-mean, L-scale, L-skewness and L-kurtosis”. We multiply this sum by the weight matrix *w*, and add the bias unit, which is [0.01 0.1 0.1 0.1 0.1 0.1], the result will be a 10,833 × 1024 matrix. The second hidden layer is calculated using a Softmax activation function. Following this, we find the average of all cross-entropies for 10,833 training images in order to learn our Softmax model, determining the weight coefficients (“regularised” weight) using gradient descent method. The learning rate (eta) is between [0.0, 1.0], and has a default value 0.01. Using parameters Iteration=500 and Cost=0.06, the prediction label is then created. The output layer is a vector of six class. Figure 5 shows an adapted MLP classifier using Softmax based gradient descent classification features using data augmentation and no data augmentation. Figure 6 shows an adapted MLP classifier using Softmax based gradient descent calculation cost and iteration and the best result is on iteration=500.

## 5. Discussion

### 5.1. Evaluation Results

The Mean Class Accuracy (MCA) is used for measuring the performance of classification classes and is adopted to score and compare the methods based on Correct Classification Rate CCR for each class. The prediction $CCR_{n}$ is calculated using two parameters from a confusion matrix, as illustrated in below equation:(16)$$CCR_{n}=\frac{1}{M_{n}}\left(TP_{n}+TN_{n}\right)$$
where $TN_{n}$ and $TP_{n}$ are the total number of True Negatives and True Positives for class *n*, respectively. $M_{n}$ represents the number of images belonging to the specfic $n_{th}$ class. The average value of $CCR_{n}$ therefore considers the *MCA* value:(17)$$MCA=\frac{1}{n}\sum_{k=1}^{n}CCR_{n}$$

Two experiments have done: In the first experiment, the testing and training datasets are randomly selected, with 2717 images used for testing and 10,833 for training. Average MCA values are 87.5% with no data augmentation, and 90.03% with data augmentation. These results show that the proposed method with data augmentation reduces the overfitting of features and results in better accuracy than the model with no data augmentation. In the second experiment, training sets have been created separately for the positive and intermediate cells. We train the same model configurations and parameters initialisation of the both categories to classify cells into positive or intermediate. we have submitted the method based on the networks error function “cross-entropy loss”. Then, the prediction label is then compared with the actual label, and the results are shown in Table 1. The results show that 3982 cell images are classified as positive and 4762 as intermediate cells. However, the difference between the values of validation and testing accuracy shows that this model may not be able to better generalise, as it sometimes suffers from overfitting to the training data, even when using the parameters chosen with the validation data. The 244,530 input features, 10,868 hidden with maximum iterations of 500 have been selected using Softmax activation to get 6 output class. “MLP performance is shown after the removal of the these features (one-by-one) at the train and test phases. If the removal of features had a negative impact on the performance of the classifier, it would be considered as highly important features for the separation of the 6 classes of HEp-2 cell”.

### 5.2. Benchmarking and Comparison with Other Techniques

Table 2 shows a comparison of the MCA values for our proposed method and other recent methods and the benchmarked dataset methods based on the Task-1 training dataset, which considered the average classification results for staining pattern images at cell level. We use the same training set of the competitions, and the results indicate that our proposed classification method outperforms all of the other methods (the first 16 rows) except the method that had done by [38] for which the result slightly higher than our proposed method. Our model achieves accuracy of 87.5% with no augmentation, and 90.03% with augmentation, and these result are higher than the best nine methods in Table 2. In addition, we implement plain MLP with the same HOS features and obtain MCA value of (84.32%. The result shows that adapted MLP is better than plain MLP.

The confusion matrix values are presented in Table 3. The confusion matrix has shown that the accuracy results of the proposed method with multi-class MLP for classification the cell into six classes are improved in comparison with values of plain MLP and another work in [7]. Furthermore, the confusion matrix values of the proposed method are different from those in [47]. For example, the speckled cell yields high accuracy of 90.00% when using the proposed method as shown in Figure 7, in contrast to the other methods, in which the features are more likely to be overlapped with the features of the centromere and homogeneous cells [47]. This difference is due to a failure to capture the cell shape information and an insufficient number of cells used to train and test resulting in the overlapped features being incorporated as true features by the proposed scenario. However, the method of research [38] results in slightly better than our proposed method by 91.02% due to discriminating between texturally similar patterns, such as Homogeneous and Speckled.

## 6. Conclusions and Future Work

In this paper, we present an effective and practical method for the classification histopathlogical images of HEp-2 staining patterns at cell level. A multi-class MLP is adapted by adding two hidden layers to automatically calculate the variation in the mean, scale, kurtosis and skewness of the input vector features based on HOS, and the Softmax regression algorithm is used to calculate the probability distribution of the variation in the mean, scale, kurtosis and skewness for each class. The neural network is trained and a predicted output vector is generated using a test dataset. The proposed algorithm is shown to perform well in practice, compared to existing methods in the literature and the state-of-the-art methods arising from benchmarked dataset. The proposed method achieved mean class accuracy of 87.50% with no augmentation and 90.03% with rotation augmentation. The performance of the proposed classification algorithm is also effective, both with and without data augmentation, and it is shown that the proposed bispectral invariant features and adapted multi-class MLP with data augmentation result in a higher classification accuracy (90.03%) than the 15 other methods reported in the two contests.

### 6.1. Proposed Methodology Advances

The results demonstrate that the proposed method is relatively invariant to the shape, rotation, scaling and shifting of cells, and is therefore robust against intra-class variation, overfitting and large-scale data volume phenomena. It has also been shown that the features are robust to variation in the mean, scale, kurtosis and skewness, and to discrimination between the different classes. The neural network is also trained to recognise the data set as positive or intermediate cells. The results demonstrate that proposed method can recognise 3982 cell images as positive from 4513, with an approximate accuracy of 88.23%, and 4762 as intermediate cell images from 5583, with an approximate accuracy of 85.30%. These results are satisfactory in comparison with the other methods investigated. The results show that the proposed method has an excellent adaptability across variations in scale, mean, skewness and kurtosis of the higher order spectra features, which is highly desirable for classification under different lab conditions.

### 6.2. Proposed Methodology Limitation

At present we have not calculated time performance, as our approach is implemented on different CPU processors and in different programming environments to other methods. Another limitation identified in the introduction is that the proposed method lose discriminative information when using fixed size images. Finally, the impact of this work is to enhance the decision of pathologists and the efficiency of CAD system, and particularly to discriminate between classes which is challenging in clinical practice. We believe that our proposed method has the potential to benefit patients for faster and more accurate diagnosis of diseases. In the future, our proposed method can be useful for designing computationally efficient HEp-2 classification method by reducing the number of network parameters ad running more data augmentation, and also designing the networks to be trained with smaller datasets. The work will study Scalability, Runtime, Memory, and Sensitivity analysis of proposed method as well as statistical analysis using T-test, as done in [48,49]. In addition, future work will study the combined Recurrent Neural Networks (RNNs) with One-vs-One classification and investigate their suitability for HEp-2 cells classification.

## Figures and Tables

**Figure 1 sensors-23-02195-f001:**
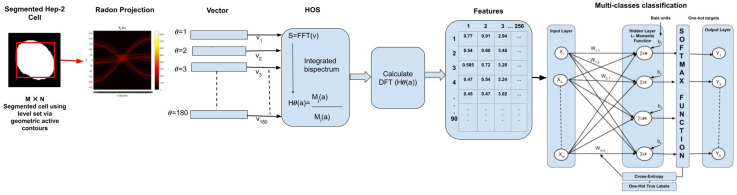
Proposed method processes to classify HEp-2 cell level patterns, including segmentation, feature extraction and multi-classes classification using adapted MLP neural network with two layers.

**Figure 2 sensors-23-02195-f002:**
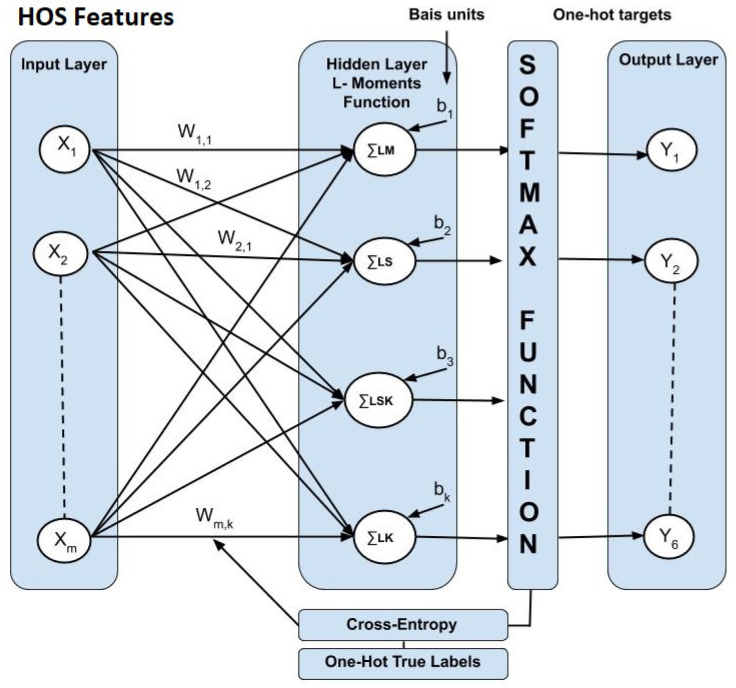
A MLP network architecture shows input layer, hidden layer and output layer.

**Figure 3 sensors-23-02195-f003:**
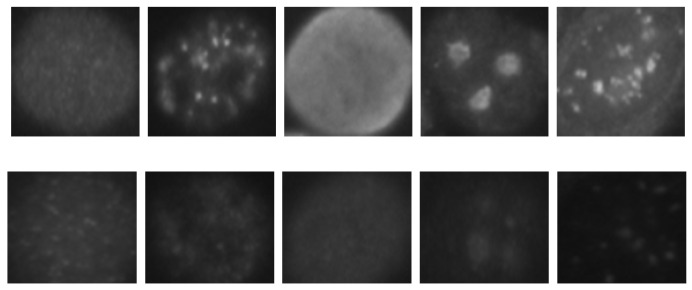
Two under-classes can be defined: (**Row 1**) positive cells (high intensity) and (**Row 2**) intermediate cells (low intensity).

**Figure 4 sensors-23-02195-f004:**
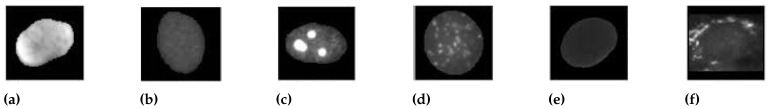
Cell staining included six pattern in the dataset: (**a**) homogeneous (Hm), (**b**) speckled (Sp), (**c**) nucleolar (Nu), (**d**) centromere (Cn), (**e**) nuclear membrane (Nm), and (**f**) Golgi (Gl).

**Figure 5 sensors-23-02195-f005:**
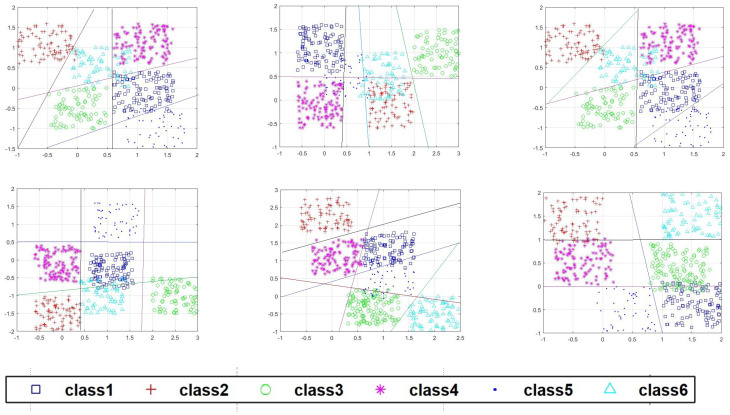
An adapted MLP classifier using Softmax based gradient descent classification features using no data augmentation (first row) and data augmentation (second row), where x-axis is training vectors, containing the number of samples and the number of features, and y-axis is the number of samples to plot decision borders.

**Figure 6 sensors-23-02195-f006:**
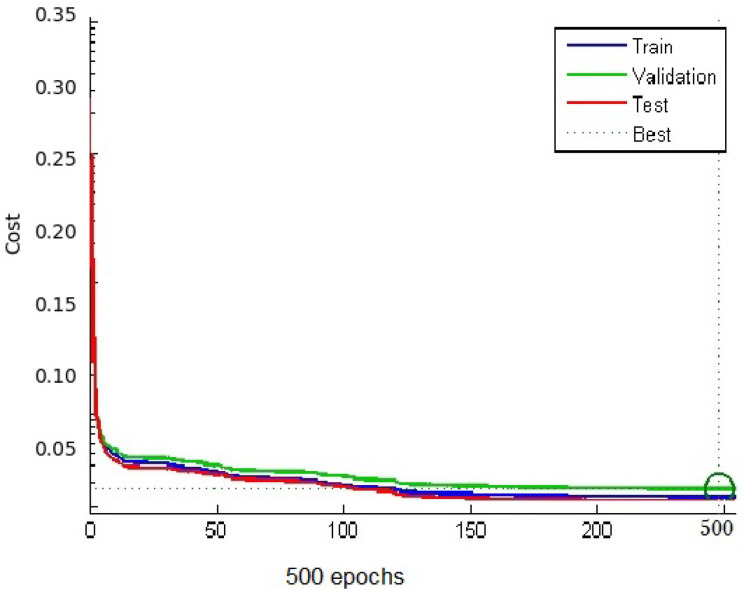
A adapted MLP classifier using Softmax based gradient descent calculation cost and iteration.

**Figure 7 sensors-23-02195-f007:**
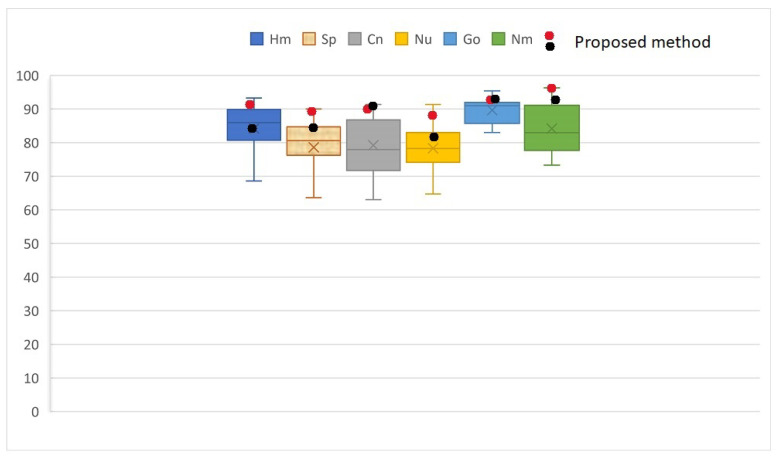
Correct classification rates (CCR) for each cell class and mean value of methods comparing with our proposed method to show the variation in classification of each class.

**Table 1 sensors-23-02195-t001:** Accuracy of positive/intermediate cells for each class and the number of images which is classified correctly from test images.

Actual Class	Positive Cell		Intermediate Cell	
	**Images**	**Rate**%	**Images**	**Rate**%
Homogeneous	722/815	88.59%	902/1055	85.50%
Speckled	900/1092	82.42%	900/1030	87.38%
Nucleola	450/500	90.00%	1000/1248	80.13%
Centromere	955/1033	92.45%	950/1022	92.96%
Nuclear Membrane	655/707	92.65%	810/948	85.44%
Golgi	300/366	81.97%	200/280	71.43%
**Overall**	**3982/4513**	**88.23%**	**4762/5583**	**85.30%**

**Table 2 sensors-23-02195-t002:** Results were achieved by the participants compared in terms of MCA to the recent methods, ICPR2014 and ICPR2016 contests (the first sixteenth rows of the table) and our adapted (MLP) classifier (the remaining rows of the table in bold text) over Task-1 training dataset.

References	Feature Extraction and Selection	Classifier	Data Augmentation	Train Set	Test Set
[30]	Trainable features	QR-FIRMLP	Mirroring and rotation	98.94	74.68
[29]	CoALBP, STR, LPC	Multiclass boosting	Rotation	100.00	81.50
[26]	SIFT and SURF with BoW	Linear SVMs & Majority voting	–	98.07	80.84
[27]	SID with soft BoW	Linear SVM (one-vs-all)	–	95.47	83.85
[33]	Trainable features	Deep CNNs	Rotation	89.02	76.26
[39]	LOAD with IFV	Linear SVM (one-vs-all)	–	99.91	84.26
[2]	Multi-resolution LP & Root-SIFT	SVMs with Platte re-scaling	Rotation	95	87.42
[40]	RICWLTP	Linear SVM (one-vs-all)	–	94.68	68.37
[41]	LCP, RIC-LBP, ELBP, PLBP, STR	SVM with Kernel RBF	Resizing & Rotation	100.00	79.91
[42]	Geometry, morphology & entropy	SVM (one-vs-one) cell level	–	90.25	80.45
[43]	Morphological & textural features	Linear SVM (one-vs-all)	–	93.82	83.06
[28]	Statistical, spectral & LDA	Gaussian mixture model	–	88.59	73.78
[24]	SIFT descriptors	Vector of hierarchically residuals	–	–	82.80
[38]	–	CNN-based Softmax	rotation, cropping & flipping	95.32	91.33
[44]	–	CNNs	–	94.01	89.52
[45]	feature concatenation & ensemble	CNNs	–	96.56	89.00
[46]	–	Very deep CNNs	Rotation		89.36
**MLP method**	**Higher order spectra**	**Plain MLP**	**No augmentation**	**90.22**	**84.32**
**Our proposed method**	**Higher order spectra**	**AMLP based L-moment**	**No augmentation**	**95.82**	**87.55**
**Our proposed method**	**Higher order spectra**	**AMLP based L-moment**	**Rotation**	**97.11**	**90.83**

**Table 3 sensors-23-02195-t003:** Confusion matrix parameters Classification results of comparable methods of ICPR2014, ICPR2016 and recent techniques using the Task-1 data set for each class.

References	Hm	Sp	Cn	Nu	Go	Nm
[30]	69.16	72.59	68.68	67.08	94.21	76.15
[29]	75.84	82.93	76.4	75.56	94.51	83.78
[26]	75.53	81.43	76.61	73.7	94.18	83.57
[27]	83.02	82.26	85.37	78.4	95.37	78.68
[33]	80.79	64.65	73.51	67.62	85.52	73.3
[39]	89.91	80.67	86.84	81.53	85.5	80.11
[2]	87.47	80.51	83.04	91.01	89.84	92.09
[40]	68.58	53.51	63.03	64.74	83.02	77.36
[41]	89.69	76.21	70.31	78.05	84.46	80.05
[42]	89.19	76.3	70.17	77.95	86.28	82.33
[43]	91.11	79.24	75.05	78.06	87.29	87.12
[28]	80.72	63.62	71.11	66.6	84.83	75.15
[24]	88.93	77.7	79.25	83.00	90.40	77.05
[38]	93.28	90.01	88.08	91.36	91.56	93.57
[44]	92.12	87.80	86.51	88.05	91.62	91.01
[45]	84.52	90.01	91.33	80.04	91.97	96.31
[46]	-	-	-	-	-	-
**plain MLP**	**82.13**	**82.14**	**90.12**	**88.13**	**90.08**	**82.10**
**Proposed method-1**	**84.53**	**85.34**	**91.32**	**80.13**	**91.80**	**91.15**
**Proposed method-2**	**91.91**	**89.81**	**89.51**	**85.97**	**91.67**	**96.16**

## Data Availability

The data is not applicable.

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
