# Peer review of "Classification of HEp-2 Staining Pattern Images Using Adapted Multilayer Perceptron Neural Network-Based Intra-Class Variation of Cell Shape"

_sensors, 2023, doi:10.3390/s23042195_

Round 1

Reviewer 1 Report

The paper is of interest and well-written. Its topic is Classification of HEp-2 Staining Pattern Images Using Adapted Multilayer Perceptron Neural Network-Based Intra-Class Variation of Cell Shape. Included are 7 figures 2 tables and 15 mathematical equations and all the relevant paper sections (including abbreviations and references). No major spelling errors were detected. Conclusions - the authors present an effective and practical method for the classification of histo-pathlogical images of HEp-2 staining patterns at the cell level. A multi-class MLP is adapted by adding two hidden layers to automatically calculate the variation in the mean, scale, kurtosis, and skewness of the input vector features based on HOS, and the Softmax regression algorithm is used to calculate the probability distribution of the variation in the mean, scale, kurtosis and skewness for each class. The neural network is trained and a predicted output vector is generated using a test dataset. The proposed algorithm is shown to perform well in practice, compared to existing methods in the literature and the state-of-the-art methods arising from the benchmarked dataset.

Author Response

The paper is of interest and well-written. Its topic is Classification of HEp-2 Staining Pattern Images Using Adapted Multilayer Perceptron Neural Network-Based Intra-Class Variation of Cell Shape. Included are 7 figures 2 tables and 15 mathematical equations and all the relevant paper sections (including abbreviations and references). No major spelling errors were detected. Conclusions - the authors present an effective and practical method for the classification of histo-pathlogical images of HEp-2 staining patterns at the cell level. A multi-class MLP is adapted by adding two hidden layers to automatically calculate the variation in the mean, scale, kurtosis, and skewness of the input vector features based on HOS, and the Softmax regression algorithm is used to calculate the probability distribution of the variation in the mean, scale, kurtosis and skewness for each class. The neural network is trained and a predicted output vector is generated using a test dataset. The proposed algorithm is shown to perform well in practice, compared to existing methods in the literature and the state-of-the-art methods arising from the benchmarked dataset.

Response to Reviewer: Thank you for appreciate comment.

There is no action required for Reviewer-1.

Reviewer 2 Report

In the manuscript Classification of HEp-2 Staining Pattern Images Using Adapted Multilayer Perceptron Neural Network-Based Intra-Class Variation of Cell Shape, Khamael Al-Dulaimi et al. proposed an adapted multilayer perceptron neural network to achieve automate HEp-2 stained cells classification. The comments are given as below.

1.      The new hidden layer in the manuscript is one innovation point of this work, but the authors did not compare the network performance of introducing L-Moment features or not. It is suggested that the author try a plain multilayer perceptron network to prove the validity of L-Moment features.

2.      The difference between the three scatter plots in each line of fig. 5 and the specific meaning and the intended problem are missing in the manuscript.

3.      Formula (6) and (7), the variable j has not been used, which means that each category j corresponding probability  is the same? Please check the formula again.

4.       in formula (12) and (14) is not indicated in the manuscript.

5.      Please indicate the meaning of RCC in Table 3.

6.      The last sentence of line 225 seems incomplete, please check it again.

7.      In line 241, the manuscript mentions that the training data is 2717 images, but in line 254, it says that the number of training images is 10833. Please check it.

Author Response

Point-1:The new hidden layer in the manuscript is one innovation point of this work, but the authors did not compare the network performance of introducing L-Moment features or not. It is suggested that the author try a plain multilayer perceptron network to prove the validity of L-Moment features.
Response-1: We test the same features using plain multilayer perceptron network and the result shows that our adapted MLP more effective than  plain MLP.  We added the below paragraph in Page (10) in Line 280  and also see  the updated Table-2 and Table-3  in Page (10-11) in the revised paper.
"In addition, we implement plain MLP with the same HOS features and obtain MCA value of (84.32\%. The result shows that adapted MLP is better than plain MLP." 

Point-2: The difference between the three scatter plots in each line of fig. 5 and the specific meaning and the intended problem are missing in the manuscript.
Response-2: We agree with the reviewer comment-2, we removed from the figure to make features of Figure.5 clearly. We add a label that mentions to  class1, class2, class3, class4, class5 and class6 of Hep-2 cell. Please see updated Figure.5 in Page(9) in the revised paper.

Point-3: Formula (6) and (7), the variable j has not been used, which means that each category j corresponding probability  is the same? Please check the formula again.
Response-3: The variable  j is used in formula 6 and it means the class while the probability  p of each class j, so we insert z of formula 7 into 6 for each class, as shown in Page(5) Line 176 in the revised manuscript.

Point-4: in formula (12) and (14) is not indicated in the manuscript.
Response-4: We revised the paper and indicated the formula (12) and (14) in Page (6)

Point-5: Please indicate the meaning of RCC in Table 3.
Response-5:  It  means Correct Classification Rate (CCR) and we explained about this measuring in Page (8) from Line 251-Line 55 in the revised paper.

Point-6: The last sentence of line 225 seems incomplete, please check it again.
Response-6: We checked it again and we rewrite the sentence to make it clear in Line 225 in Page (8) the revised paper.
" We obtained a set of 23,040 features for each image.”

Point-7: In line 241, the manuscript mentions that the training data is 2717 images, but in line 254, it says that the number of training images is 10833. Please check it.
Response-7 : We agree with reviewer comment. It is an accidentally mistake and we revised the manuscript. The testing and training datasets are randomly selected, with 2, 717 images used for testing and 10, 833 for training.

The grammar and proofreading have been done to the paper by professional English services.

Reviewer 3 Report

The Abstract is suggested to be improved where the contribution and findings of the work should be highlighted.

How to tune MLP hyperparameters and their value
What are MLP configurations parameters
The authors should specify more details regarding the Experiment for the proposed algorithm.
The authors should provide more details regarding the analysis of the results.

The advantage and disadvantages of the work are suggested to be highlighted in comparison with extant studies or methods.
How to initialize the agents in the proposed Algorithm?

Some additional experiments are required:
a. - Scalability
b. - Runtime
c. - Memory
d. - Sensitivity analysis
Read and cite these references.

Statistical analysis should be carried out to demonstrate that the experimental results are significant. Such as the ANOVA test and T-test
Read and cite these references.
E.-S. M. El-kenawy, H. F. Abutarboush, A. W. Mohamed and A. Ibrahim, “Advance artificial intelligence technique for designing double T-shaped monopole antenna,” Computers Materials & Continua, vol. 69, no. 3, pp. 2983-2995, 2021.

Some syntax errors or improper expressions exist in the manuscript.

More up-to-date studies are suggested to be cited.

Author Response

Response to Reviewer 3 Comments

Point 1: The Abstract is suggested to be improved where the contribution and findings of the work should be highlighted.

Response 1: The abstract has been improved. We highlighted the paragraphs with contribution and findings of the work. The first paragraph is bout contribution, the second is about objective and the final is about findings of the work.

Author action:

In this paper, a multi-class multilayer perceptron technique is adapted by adding a new hidden layer
to automatically calculate the variation in the mean, scale, kurtosis and skewness of higher order
spectra features of the cell shape information. The adapted technique is then jointly trained and the
probability of classification calculated using a Softmax activation function.

This method is proposed to address overfitting, stained and large-scale data volume problems, and
classify HEp-2 staining cells into six classes.

An extensive experimental analysis is studied to verify the results of the proposed method. The
technique has been trained and tested on the dataset from ICPR2014 and ICPR2016 competitions and
the experimental results have shown that the proposed model achieved higher accuracy of 90.3%
(with data augmentation) than of 87.5% (with no data augmentation). In addition, the proposed
framework is compared with state-of-the-art of HEp-2 cell image classification models , as well as
the results the of ICPR2014 and ICPR2016 competitions using the Task-1 training dataset.The results
demonstrate that our proposed method effectively outperforms recent methods.

Point 2: How to tune MLP hyperparameters and their value

Response 2: We provided with MLP hyperparameters and their value.

Point 3: What are MLP configurations parameters

Response 3: We added MLP configrations parameters, including input layer, hidden layer, weights, learning rate, iteration and output layer in page (8) in Section 4-2 Implementation of proposed method in Point-3. Below some details.

The input layer is a vector of (23,040) features multiplying by training images size of 10, 833 × 23, 040. Then, we  initialise the parameter of weight matrix size of 10, 833 × 23, 040 × 6 ( one column

for each class and one row feature), where k represents four weights for each node.  For example, the first row the matrix of dimensional weights is [0.1 0.2 0.3 0.4 0.5 0.6]. We construct a neural network with two hidden layers. The first hidden layer is  calculated by summing the L-moments function, which includes the L-mean, L-scale, L-skewness and L-kurtosis. We multiply this sum by the weight matrix w, and add  the bias unit, which is [0.01 0.1 0.1 0.1 0.1 0.1], the result will be a 10, 833 × 1024 matrix. The second hidden layer is calculated using a softmax activation function.  Following this, we find the average of all cross-entropies for 10, 833 training images in  order to learn our softmax model, determining the weight coefficients (“regularised” weight) using gradient descent method. The learning rate (eta) is between [0.0, 1.0],  and has a default value 0.01. Using parameters Iteration = 500 and Cost = 0.06, the prediction label is then created.

Point 4: The authors should specify more details regarding the Experiment for the proposed algorithm.

 Response 4: We added more details regarding the Experiment for the proposed algorithm and explained about the process in Page(8) in Section 4.2 as points which including pre-processing, feature extraction and adaptive MLP.

Point 5: The authors should provide more details regarding the analysis of the results.

Response 5: We povided more details regarding the analysis of results and explained it in Page(8-9) in Section discussion from Line 253-271 in the revised paper and highlighted  in Blue.

Point 6: The advantage and disadvantages of the work are suggested to be highlighted in comparison with extant studies or methods

Response 6: We discussed the advantage and disadvantages of the propsed method  in Sub-section 6-1 and 6-2, respectively from Line 311 to Line 337 .This discussion includes the ability of the proposed method in comparsion with other existing methods.

Point 7: How to initialize the agents in the proposed Algorithm?

Response 7:

Point 8: Some additional experiments are required:

  1. - Scalability
  2. - Runtime
  3. - Memory
  4. - Sensitivity analysis

Read and cite these references.

Statistical analysis should be carried out to demonstrate that the experimental results are significant. Such as the ANOVA test and T-test

Read and cite these references.

E.-S. M. El-kenawy, H. F. Abutarboush, A. W. Mohamed and A. Ibrahim, “Advance artificial intelligence technique for designing double T-shaped monopole antenna,” Computers Materials & Continua, vol. 69, no. 3, pp. 2983-2995, 2021.

Response 8: We have done statistical analysis based on Mean Class Accuracy (MCA) using Correct
Classification Rate CCR for each class because most techniques related to Hep2-cell classification are used the measuring. This helps to compare with those techniques using the same measuring to get banchmarking. Also, it equires time to read and  implement thsese experiments. However,  the additional experiments  and analysis can be done as a future work. A sentence about this analysis and references  has been provided  in Conclusion Section in Page-12 from Line 335 to Line 336.

Point-9 More up-to-date studies are suggested to be cited.

Response 9: we updated the references  in the revised paper.

Note:The grammar and proofreading have been done to the paper by professional English services.

Round 2

Reviewer 3 Report

Some syntax errors or improper expressions exist in the manuscript.
More up-to-date studies are suggested to be cited.

Author Response

Response to Reviewer 3 Comments

Point 1: Some syntax errors or improper expressions exist in the manuscript.

Response 1:  We did proofread and editing using Track changes function to show that we fixed Some syntax errors or improper expressions.

Point 2: More up-to-date studies are suggested to be cited.

Response 2: we have added up-to-date studies and suggested more references paper.

[4] CRUVINEL, W. D. M., Andrade, L. E. C., MUHLEN, C. A. V., Dellavance, A., Ximenes, A. C., Bichara, C. D., ... & Francescantonio, P. L. C. (2022). V Brazilian consensus guidelines for detection of anti-cell autoantibodies on hep-2 cells . Advances in Rheumatology.

[4] Srinidhi, C. L., Kim, S. W., Chen, F. D., & Martel, A. L. (2022). Self-supervised driven consistency training for annotation efficient histopathology image analysis. Medical Image Analysis, 75, 102256.

[7]Zhou, X., Li, Z., Xue, Y., Chen, S., Zheng, M., Chen, C., ... & Tong, T. (2023). CUSS-Net: A Cascaded Unsupervised-based Strategy and Supervised Network for Biomedical Image Diagnosis and Segmentation. IEEE Journal of Biomedical and Health Informatics.